# The Degree of Acceptance of Cocoon Strategy of Vaccination against Influenza and COVID-19 in Palliative Home Care Professionals and Caregivers

**DOI:** 10.3390/vaccines11071235

**Published:** 2023-07-12

**Authors:** Agnieszka Kluczna, Rafał Orzeł, Anna Bardowska, Tomasz Dzierżanowski

**Affiliations:** 1Department of Palliative Medicine, Institute of Sciences, University of Opole, 45-052 Opole, Poland; agnieszka.kluczna@uni.opole.pl; 2Laboratory of Palliative Medicine, Department of Social Medicine and Public Health, Medical University of Warsaw, 02-007 Warsaw, Poland

**Keywords:** cocoon strategy, vaccination, influenza, COVID-19, palliative care

## Abstract

Palliative care patients are an immunocompromised population, so the cocooning strategy of vaccinating those around them is a suitable protective strategy against infections. This is especially significant for infectious diseases such as influenza and COVID-19, which pose a challenge to the healthcare system. In order to improve the patient’s quality of life, it is necessary to develop research-based, defined strategies. This questionnaire-based study was conducted to determine the level of knowledge about influenza and SARS-CoV-2 coronavirus vaccination among the factual caregivers and medical staff in the palliative care setting. The survey revealed that the sources of knowledge about vaccination varied and depended on one’s role. Doctors and nurses used professional literature, while other medical professionals relied on the Internet, mass media, and information from family and friends. The study also showed that adherence to vaccination guidelines was not associated with COVID-19 incidence. The overall opinion on vaccination was positive, but the degree of acceptance varied by the role. Palliative care nurses and caregivers were the groups that were the least accepting of vaccination. To improve the acceptance of vaccinations, a remedial program based on professional education should be implemented using the sources declared by the respondents. It may help improve the quality of life for palliative care patients and prevent the spread of infectious diseases.

## 1. Introduction

Influenza viruses are causing seasonal epidemics, regularly occurring in the Northern and Southern Hemispheres each winter and resulting in 3 to 5 million cases and 290 to 650 thousand respiratory deaths annually. The leading group of influenza deaths is older adults in high-income countries and children under five in developing countries [1]. The threat of the next influenza pandemic was outclassed by the coronavirus disease 2019 (COVID-19) pandemic, as announced by WHO Director-General on 11 March 2020 [2]. More than 663 million cases of COVID-19 cases, including 6707 million deaths, were reported by 18 January 2023 [3]. WHO documented the emergence of new variants of SARS-CoV-2 in collaboration with national authorities, institutions, and researchers. More than ten lineages of SARS-CoV-2 were assessed and designated in January 2023 as they were detected from November 2022 to January 2023 [4].

Both of the viral infections, influenza and COVID-19, are airborne diseases with similar symptoms [1,5]. Comparative analysis of almost 4000 COVID-19 patients in the United States (US) vs. almost 5500 influenza patients in the US indicated a higher risk of serious complications and death in the case of SARS-CoV-2 infection [6]. A severe COVID-19 course was connected with a five times higher case–fatality ratio compered to severe influenza infection (21.0% vs. 3.8% respectively) [6].

The risk of infection and vaccines’ immunogenicity, efficacy, and effectiveness depend on the patient’s health status. Palliative care patients are affected by severe chronic conditions, such as cancer, end-stage organ failure, or several neurologic diseases; many are older adults. They have impaired humoral and cellular immunity and are more susceptible to infectious agents. Their immunological response to vaccines is expected to be less effective than that of healthy people [7]. Antibody levels considered protective in healthy individuals may not prevent clinical infection in malignant disease patients [8]. The results indicate that influenza vaccination is probably effective and can be offered to terminally ill cancer patients with a life expectancy of about three months in a home care palliative care unit [9]. Vaccination of the remaining persons in close home contact with patients who require palliative care becomes particularly important in reducing the transmission of pathogens (and thus reducing the risk of infection in the patient), which is the basis of the cocooning strategy of vaccination [10]. It aims to protect people who cannot be vaccinated due to medical contraindications by vaccinating persons prone to illness from their immediate environment, such as family or caregivers [11]. A cocoon strategy has been used to protect newborns from pertussis by vaccinating their parents and oncological patients and their first household contacts against influenza [12,13]. Influenza vaccination of healthcare workers in elderly-care hospitals was associated with decreased mortality among patients [14]. Application of cocoon strategy has also been suggested regarding COVID-19. Since households remain one of the most common transmission sources, vaccinating caregivers may help protect oncological patients [15].

This study aimed to determine attitudes towards vaccinating against two epidemiologically critical infectious diseases, namely influenza and COVID-19, presented by palliative home care professionals and caregivers. The results can be used to design educational materials promoting cocoon vaccinations, given that further pandemics are likely, and the learning points from the COVID-19 pandemic should be implemented in advance.

## 2. Materials and Methods

### 2.1. Data

We performed a questionnaire-based survey of palliative home care providers in Poland on 9 February–28 March 2021. The questionnaire, designed by the authors, consisted of forty-two questions and claims in Polish as a native language. It comprised general statements about vaccination without introducing the concept of the cocoon strategy to avoid bias. After the first twenty questionnaires were collected, it was validated (preliminary testing), and then the invitation to the survey was sent directly to palliative home care centers throughout Poland. The respondents voluntarily and anonymously filled in an Internet (Google Forms) or paper form, depending on their preferences, with no exclusion criteria applied. The survey was conducted according to the guidelines of the Declaration of Helsinki after prior approval by the Bioethics Committee of the Medical University of Warsaw, Poland (AKBE/21/2021 of 2021-02-08).

We used COVID-19 as a synonym for SARS-CoV-2 infection, because it was better understood and recognizable. For further analysis, a 5-point ordinal scale was used for Likert-scale questions (strongly disagree—1, disagree—2, neutral—3, agree—4, strongly agree—5). A native speaker validated an English adaptation of the questionnaire for reproducibility in other than Polish settings (Appendix A).

The results were analyzed in demographic subgroups of role, age, gender, marital status, education, employment status, and rural/urban location of the practice.

### 2.2. Sample Size

In January 2021, the National Health Fund concluded 382 contracts in Poland who provide specialist palliative home care services and 150 contracts with palliative care inpatient centers [16]. Almost 3100 nurses work in palliative home care in Poland (NFZ, the National Consultant’s registry of December 2020). The total number of physicians is unknown, but it is likely between 1000 and 1800 [16]. There is also no publicly available information on the numbers of psychologists (over 700), physiotherapists (over 1000), and employed non-medical caregivers. Therefore, we planned for at least 400 respondents, which is over the recommended minimum of 384 for the unknown population size.

### 2.3. Statistical Analysis

In descriptive statistics, mean values and 95% confidence intervals (95 CI) were applied for continuous variables; medians and quartile values (Q25, Q75) for ordinal variables; and percent values for nominal variables. The normality of distribution was tested using the Shapiro–Wilk test. Non-parametric (Kruskal–Wallis and Mann–Whitney U) tests and Spearman rank-order correlations were used for ordinal variables, and Pearson’s chi-squared test for nominal variables. The Kruskal–Wallis test was used for non-normally distributed continuous variables (age). The Wilcoxon signed-rank test was used to compare measurements on the same individuals (e.g., opinion on influenza and COVID-19 vaccination safety).

A *p*-value < 0.05 was considered statistically significant with Bonferroni correction where appropriate. All statistical analyses were performed with Statistica 13—TIBCO Software Inc. (2017). Statistica (data analysis software system), version 13. http://statistica.io (accessed on 20 May 2022).

## 3. Results

### 3.1. Demographic Data

We collected 450 web-based and paper-form questionnaires and excluded 13 apparent duplicates. A total of 437 respondents were included in the analysis. Additionally, we received 19 questionnaires after the data collection deadline—these were not included in the analysis but quality checked and added to the database for any future cumulative analysis or comparisons. 

Table 1 presents the demographic characteristics of the sample. The majority (85%) of respondents were women. The respondents’ mean age was 49 (95% CI 48–50), and the youngest and oldest professionals were 21 and 83 years old, respectively.

There were statistically significant differences in age between the role groups. The oldest appeared to be nurses—mean 52.6 (95% CI 51.5–53.8) and caregivers—51.6 (95% CI 48.0–55.1), who were significantly older than physicians—46.5 (95% CI 44.3–48.7), and the youngest: psychologists—39.4 (95% CI 36.1–42.6) and physiotherapists—39.1 (95% CI 35.6–42.6) years.

The sample is representative of palliative home care professionals and caregivers in Poland.

### 3.2. Source of Information on Vaccines

The source of information on vaccines differed, depending on the professional role (Figure 1). The primary sources for physicians were medical journals, to which they referred most often (77% of respondents), other medical professionals (67%), and the Internet (46%). Other PC professionals referred to medical professionals as the primary source of information (caregivers 83%, physiotherapists 74%, nurses 69%, and psychologists 57%). Media such as television, radio, press, and the Internet were similarly essential for them. Nurses also used medical journals comparably often (47%). The differences between role groups were statistically significant for medical journals (*p* < 0.00001), mass media (TV, radio, press; *p* < 0.00001), social web portals (*p* = 0.03), and family/acquaintance (*p* = 0.0008) as sources of information on vaccines.

Age did not determine the primary sources of information, except for the respondents under 35 years, who referred to family and acquaintances much more often than others (33%; *p* = 0.0002).

Women more often declared mass media (45% vs. 28%; *p* = 0.01) and less often medical journals (48% vs. 67%; *p* = 0.005) than men as sources of information.

Education impacted the choice of source of information. The higher the education grade, the more often medical journals and the less frequently media and family were used. 

The geographical location of practice, marital status, and employment status did not affect the use of different sources of information.

### 3.3. Observing Epidemic Safety Regulations on Personal Protective Measures (PPMs)

Almost all palliative care professionals and caregivers declared disinfecting hands (99%), wearing a mask when approaching the patient (99%), and maintaining a social distance (94%), except two nurses and one caregiver. No statistical differences between groups were observed.

A face shield or helmet was used when approaching the patient by 38% of professionals, with no statistical difference between roles, except for caregivers, who used them less often than the others (12%; *p* = 0.006).

Observing epidemic safety restrictions did not differ between age, gender, marital status, education, rural/urban location, or employment groups. It was not associated with a kind of information source, except for wearing a face shield/helmet, which was more frequent when a medical professional was a declared source of information (47% vs. 33%; *p* = 0.008). Some impacts of the Internet and social media on maintaining social distance have been observed (*p* = 0.042; statistically insignificant when applying the Bonferroni correction for multiple comparisons).

Neither history of own suffering from COVID-19 nor being in the nearest vicinity was associated with observing safety restrictions. More respondents wore a face shield/helmet when they had previously experienced death from COVID-19 in the immediate vicinity (*p* = 0.022).

### 3.4. General Opinion on Vaccinations

General opinions on vaccinations were positive (Figure 2), with significant differences between the groups for each question (Appendix A).

Physicians, almost unanimously, strongly agreed that (1) thanks to vaccinations, many dangerous infectious diseases do not occur today, (2) vaccinations are the most effective way to protect against infectious diseases, and (3) they provide more benefits than risks. They strongly disagreed that vaccinations are unnecessary because infectious diseases are rare, and that they do not need to vaccinate because of the low risk of getting infected (Appendix A). They also disagreed to the highest level that vaccinations are promoted not because they are really needed, but because it is in pharmaceutical companies’ interests, although the response was not unanimous.

The opinions of the other role groups for each question were generally less favorable, as presented in Appendix A. While all physicians were convinced that “thanks to vaccinations, many dangerous infectious diseases do not occur today,” this opinion was shared by 96% of psychologists, 95% of physiotherapists, 92% of nurses, and only 65% of caregivers (*p* < 0.0001).

Unlike physicians, significantly more respondents in other role groups claimed that “vaccinations are unnecessary because infectious diseases are rare,” especially among caregivers and nurses (*p* < 0.0001).

“Vaccinations provide more benefits than risks” was claimed by 95% of physicians, 92% of psychologists, 90% of physiotherapists, 88% of caregivers, and 82% of nurses (*p* < 0.0001).

The persons who opposed the statement “I do not need to vaccinate because the risk of getting sick and infecting my patients is small, as I respect the rules of isolation and have no contact with other people,” accounted for 86% of physicians, 75% of psychologists, 68% of nurses, 65% of physiotherapists, and 59% of caregivers (*p* < 0.0001).

All opinions appeared well correlated with each other, with logical direction (Appendix A).

### 3.5. Attitude toward Influenza and COVID-19 Vaccination

Figure 3 illustrates the distribution of responses to questions about influenza and COVID-19 in the context of vaccination. Respondents regarded influenza vaccination as safe (84%) and effective (76%), while for COVID-19 vaccination, these figures were lower (71% and 63%, respectively; *p* < 0.00001). The two vaccinations were perceived differently concerning safety (the Wilcoxon test *p* < 0.00001), effectiveness (*p* = 0.00003), and the necessity to avoid infection (*p* = 0.00001).

The degree of conviction to vaccinate against COVID-19 to be safe or ensure such safety for the patient appeared much higher than that against influenza (82% vs. 60% and 84% vs. 58%, respectively; *p* < 0.00001). Likewise, the conviction that “chronic diseases cause a severe course of infection” and “the infection is dangerous for the patient” was stronger for COVID-19 than for influenza (*p* < 0.00001).

The declared knowledge of vaccination against the two infections seemed similar, although statistically better for influenza (*p* = 0.000009).

There were few statistically significant differences between role subgroups when applying the Bonferroni correction for multiple comparisons (α = 0.0019), as detailed in Appendix A. Rural and small-town professionals did not see a need for vaccination due to the low risk of getting infected to a more significant extent than those working in urban centers (*p* = 0.0007). The opinion that vaccinations are unnecessary because infectious diseases are rare was more frequent in persons in retirement than those working or studying (*p* < 0.0001). Persons older than 65 years were more likely to claim that vaccinations are unnecessary because infectious diseases are rare (*p* = 0.0004); however, a post hoc analysis showed that this finding should be taken cautiously, as the representation of nurses in this subgroup was predominant. Other demographic factors did not correlate with any opinion on influenza or COVID-19 vaccinations.

Respondents who were infected by COVID-19 were less convinced that COVID-19 vaccination is safe (*p* = 0.0003), effective (*p* = 0.0043), or necessary to avoid infection (*p* = 0.0003) than those who did not. They were also less likely to get vaccinated to ensure patient’s or their own safety (*p* = 0.0007 and *p* = 0.0023, respectively). The experience of COVID-19 infection in the immediate vicinity was not associated with more favorable opinions about vaccination against it. However, professionals who experienced death due to COVID-19 in the immediate vicinity were more likely to get vaccinated against it to ensure their own epidemiological safety (*p* = 0.0001).

Despite the observed differences, all opinions about influenza and COVID-19 appeared strongly positively correlated with each other (Appendix A). The general opinions on vaccinations also correlated with those on influenza and COVID-19 vaccinations (Appendix A).

### 3.6. Undergoing Vaccination against Influenza and COVID-19

#### 3.6.1. Influenza Vaccination

Regularly undergoing influenza vaccination every year was declared by 37% of physicians, 10% of nurses and caregivers, 6% of physiotherapists, and 4% of psychologists. Never being vaccinated against influenza was reported by 21% of physicians, 65% of physiotherapists, 64% of caregivers, 54% of psychologists, and 53% of nurses. The differences were statistically significant (*p* < 0.00001). Men more often get vaccinated against influenza regularly every year (31% vs. 15%; *p* = 0.002). Other demographic factors correlated insignificantly.

The general attitudes towards vaccinations, as well as those specifically regarding influenza and COVID-19, correlate with undergoing vaccination. Results using an ordinal scale for the question “Do you get vaccinated against influenza?” (I have never got vaccinated—0, only this season—1; I got vaccinated, but not regularly—2; regularly every year—3) appeared well correlated with positive opinions about the vaccination’s safety (r = 0.49), effectiveness (r = 0.46), necessity (r = 0.5), and declared willingness to get vaccinated in order to ensure their own (r = 0.62) or their patient’s epidemiological safety (r = 0.58); *p* < 0.0017.

#### 3.6.2. SARS-CoV-2 Vaccination

An experience with COVID-19 impacted the willingness to get vaccinated. Significantly more respondents who had not been infected by COVID-19 declared undergoing SARS-CoV-2 vaccination (95% vs. 78%; *p* < 0.00001). Likewise, 88% of professionals who experienced COVID-19 in their immediate vicinity were likely to get vaccinated versus 78% of those who did not (*p* = 0.023).

SARS-CoV-2 vaccination was already administered or scheduled for 94% of physicians, 87% of physiotherapists, 83% of caregivers, 82% of nurses, and 79% of psychologists (*p* = 0.00001). Only 4% of physicians will not get vaccinated to protect their patients. However, they indicated personal health issues as a contraindication for any vaccinations. On the contrary, 11% of nurses, 9% of physiotherapists, 10% of caregivers, and 8% psychologists did not intend to get vaccinated, shared their doubts about the vaccination’s safety or necessity, and did not plan to get vaccinated. Additionally, 5% of respondents will not get vaccinated because of declared contraindications or currently high antibody titer. No demographic factor appeared associated with readiness to get vaccinated.

The willingness to vaccinate against COVID-19 correlates with the degree of conviction about the vaccination’s safety (r = 0.46), effectiveness (r = 0.41), the necessity to avoid complications (r = 0.47), and declared intention to get vaccinated in order to ensure their own (r = 0.66) or their patient’s epidemiological safety (r = 0.63); *p* < 0.0017.

Almost 24% of all respondents reported any allergic reaction in the past. The ratio of persons undergoing COVID-19 vaccination was lower in this group (79% vs. 88%; *p* = 0.048). No such difference was observed concerning influenza vaccination.

## 4. Discussion

Our report is the first, and to the best of our knowledge, the only one, relating to health professionals in Poland. The survey was conducted almost a year after the COVID-19 outbreak, at its peak, shortly after vaccines were accessible for free to all medical professionals, with no exclusions nor limitations, in Poland. The cocoon is a basic strategy to protect the immune-impaired population. As we faced reports on several cases of hesitance towards vaccines among palliative care professionals, we wanted to examine the degree of their willingness to get vaccinated against COVID-19 and influenza, and thus a possible impediment to the cocoon strategy.

Nurses, unsurprisingly, were the largest occupational group we surveyed in our survey—they accounted for half of the total respondents, followed by physicians. The sample group is representative of the medical professionals working in the home and inpatient palliative care centers, considering age and sex.

In general, the professionals used self-protection measures when contacting patients. However, disappointingly, the minority used face shields—in particular, few medical caregivers did so. Such an attitude confirmed our suspicions of unsatisfactory awareness of the importance of PPM. This issue should be repeatedly addressed in awareness-raising programs for all professions repeatedly.

Most respondents declared they would get vaccinated against COVID-19 (82%) and influenza (60%). Our findings are consistent with studies from Germany [17], Italy [18], and France [16]. Priority vaccination of medical employees providing services in inpatient medical centers, nursing homes, and patient homes for older adults and disabled people has been implemented as national programs in many countries, including Poland.

Physicians appeared the most convinced about the safety, effectiveness, and necessity of both influenza and SARS-CoV-2 vaccinations from among all medical professionals. They were also ready to get vaccinated for their own safety to the highest degree. On the contrary, nurses and caregivers appeared least convinced and least likely to get a vaccination for their own protection. The fraction of nurses declaring undergoing SARS-CoV-2 vaccination (79%) is, for example, similar to that in France (76.2%) [16] and lower than that in Italy (91.5%) [19] or Germany (91%) [17]. The rapid pace of vaccine development has contributed to their low acceptance, fueled by concerns about their safety and efficacy [20].

Likewise, physicians intended to be vaccinated against influenza for the patients’ safety to the highest degree (91%), while nurses and caregivers were much less often. It is unclear whether it is lower awareness of the cocoon strategy or a more general problem of decreased responsibility for patients that is the reason for the poorer results among nurses. In addition, the two causes above may overlap with each other. Our results fit those from European Countries, the US, China, and Turkey [21]. Vaccine acceptance rates were the highest among physicians and lowest among nursing professionals, such as registered nurses, nursing associates, or auxiliary nurses within Europe [21].

The results for influenza follow the pattern for COVID-19, although, except for physicians, fewer medical professionals would get vaccinated for patients’ protection. Only 10% of nurses and caregivers in our survey declared regular vaccination in the previous years. One of the reasons for this may be the cost of vaccines, as they remain unreimbursed. However, certified nurses and caregivers likely need more evidence-based training on vaccines. In a recent review (not including Poland), healthcare workers with a history of previous influenza vaccination were more likely to receive the COVID-19 vaccine [21].

The group of people most hesitant to vaccinate was nurses who had not recently been vaccinated against influenza [22]. Vaccination beliefs are partly determined by individuals’ values on the risks and benefits of COVID-19 vaccines. Factors positively influencing vaccination uptake include psychological conditions such as fear of COVID-19 and increased disease susceptibility [23]. The worldview associated with non-acceptance of vaccination is related to a lack of faith in vaccines, previous history of infection, and a lack of knowledge [21]. The highest COVID-19 vaccination uptake was found in an Italian cross-sectional study (98.9%) [24].

The source of information on vaccine safety varied depending on sex and education grade. Medical journals were the primary source of information for physicians, unlike Southern California physicians who obtained vaccination knowledge from government agencies and their employers [25]. Women and less educated respondents chose mass media more often than medical journals. It should be noted that misinformation has been present on the Internet, including social media, on how vaccines are developed and distributed [20]. Age did not determine the primary sources of information, except for the responders under 35 years. More often than older respondents, young healthcare workers asked their family members for vaccine information.

Physicians, on the other hand, were strongly in agreement that vaccinations are very effective in preventing contagious diseases, and thanks to them, many dangerous diseases do not occur today. On the contrary, nurses less often considered vaccines necessary due to their beliefs that infectious diseases are uncommon presently. Moreover, our study revealed concerns about vaccination safety—only 82% of nurses replied that vaccinations provide more benefits than risks compared to 95% of physicians. This is consistent with other studies, according to which side effects are the main reason for vaccine hesitancy [22,26,27,28].

Detailed data identifying barriers among medical staff are representative and allow for the development of “tailor-made” educational programs dedicated to opponents of preventive vaccination. Based on the research results, it is also possible to plan the forms and sources of information, i.e., materials based on scientific reports for professionals and reliable media coverage for caregivers. This can only be achieved by understanding and responding appropriately to beliefs, concerns, and expectations, mainly through 5C Model—Confidence, Constraints, Complacency, Calculation, and Collective—as a framework to understand these concerns and develop strategies to increase vaccination coverage in healthcare workers [29]. Actions may be related to constructing a chatbot that allows discussion and dialogue with people about COVID-19 vaccination concerns by presenting non-judgmental scientific evidence. Changing attitudes is essential in the face of potentially upcoming pandemics. Remedial activity is a subsequent step following the research conducted by the authors.

## 5. Conclusions

More healthcare professionals got vaccinated against COVID-19 than influenza. Only a small amount of healthcare professionals get vaccinated against influenza regularly every year. Certified nurses and caregivers showed the lowest willingness to be vaccinated. Thus, campaigns to raise awareness or training addressed explicitly to this professional group would be of value. Due to the most frequent contact with patients, nurses are one of the most-trusted professional groups. Therefore, increasing their medical knowledge on preventive strategies is essential for improving patients’ trust in vaccinations. There is a need for evidence-based sources of knowledge that are easily accessible in the media, as many respondents declared using the Internet, press, or television to seek information related to vaccines. Local initiatives should be developed to improve awareness and acceptance of preventive vaccinations. Medical professionals’ attitudes toward vaccination require further research that considers the change in beliefs over time and the factors that influence the formation of attitudes toward vaccination. By taking care of current vaccinations, we can protect the entire society.

## 6. Limitations

The study’s limitations are the rapid changes in the area we are trying to define. The media activity of opponents of vaccination and the state of the pandemic, which also influenced the formation of attitudes, were also significant in shaping the opinions of medical workers and caregivers.

## Figures and Tables

**Figure 1 vaccines-11-01235-f001:**
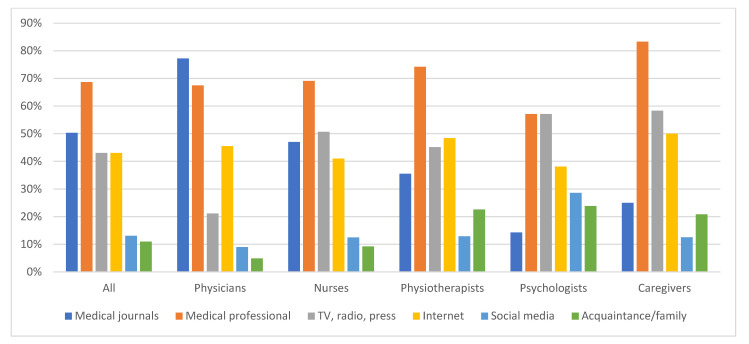
Source of information on vaccines.

**Figure 2 vaccines-11-01235-f002:**
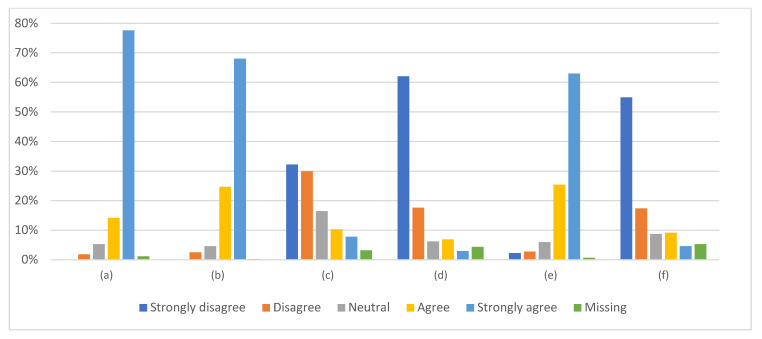
General opinion on vaccines and vaccination: (**a**) Thanks to vaccinations, many dangerous infectious diseases do not occur today. (**b**) Vaccinations are the most effective way to protect against infectious diseases. (**c**) Vaccinations are promoted not because they are really needed but because it is in pharmaceutical companies’ interests. (**d**) Vaccinations are unnecessary because infectious diseases are rare. (**e**) Vaccinations provide more benefits than risks. (**f**) I do not need to vaccinate because the risk of getting sick and infecting my patients is small, as I respect the principles of isolation and have no contact with other people.

**Figure 3 vaccines-11-01235-f003:**
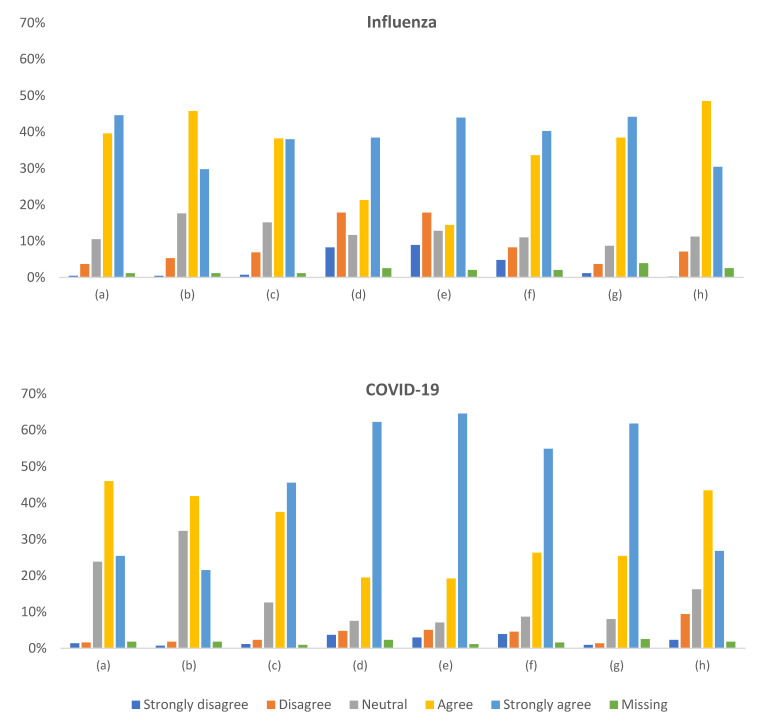
Opinion on influenza and COVID-19 in the context of vaccination: (**a**) Vaccination against this infection is safe. (**b**) Vaccination against this infection is effective. (**c**) Vaccination against this infection is necessary to avoid illness, complications, and hospitalization. (**d**) I will get vaccinated against this infection to ensure my patients’ epidemiological safety/charge. (**e**) I will get vaccinated against this infection in order to ensure my own epidemiological safety. (**f**) Chronic diseases (e.g., cancer, lung and circulatory diseases, diabetes, obesity) cause a severe course of this infection. (**g**) This infection is dangerous for my charge/patients. (**h**) My knowledge of vaccination against this infection is sufficient.

**Table 1 vaccines-11-01235-t001:** Sample characteristics (N = 437).

Characteristic	N	%
**Gender**		
Female	372	85%
missing data	4	1%
**Age**		
<35 years	48	11%
35–50 years	165	38%
50–65 years	174	40%
>65 years	32	7%
missing data	18	4%
**Location of the PHC center**		
rural area	26	6%
town up to 50,000 inhabitants	106	24%
city 50,000–100,000 inhabitants	93	21%
city 100,000–500,000 inhabitants	143	33%
city >500,000 inhabitants	64	15%
missing data	5	1%
**Marital status**		
single	110	25%
couple	315	72%
missing data	12	3%
**Education**		
primary	1	0.2%
basic vocational	10	2%
high school	81	19%
bachelor’s degree	72	16%
university degree	273	62%
**Employment status**		
studying and working	19	4%
employed only	380	87%
working on retirement/pension	38	9%
**Role**		
physician	123	28%
nurse	217	50%
psychologist	24	5%
physiotherapist	31	7%
medical caregiver	42	10%

## Data Availability

The data presented in this study are available on request in a form of Excel spreadsheet or a flat file from the corresponding authors for any academic use upon citation of this article.

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
