# Peer review of "The Degree of Acceptance of Cocoon Strategy of Vaccination against Influenza and COVID-19 in Palliative Home Care Professionals and Caregivers"

_vaccines, 2023, doi:10.3390/vaccines11071235_

Round 1

Reviewer 1 Report

The article is interesting. Some comments for authors response

1. Authors may add more information in the introduction about palliative care patients and their characteristics to support the aim of the study.

2. Please specify the category of patients and caregivers, are the PC centers provide hospice services or general palliative support

3. Quetions of the questionnaire seem more general to vaccination and not to cocoon strategy: please clarify

4. The population selected has some subtypes which may necessitate more sample size.

Author Response

Thank you for your invaluable remarks and suggestions. We followed and implemented them thoroughly. Please, find our responses below.

1. Authors may add more information in the introduction about palliative care patients and their characteristics to support the aim of the study.

Response: We added a sentence that could clarify who palliative care patients are. Moreover, following the instructions of the other Reviewer, we shortened the “Introduction” chapter significantly, focusing on the paper’s goal.

2. Please specify the category of patients and caregivers, are the PC centers provide hospice services or general palliative support

Response: We did not survey patients in this research paper. We added a description: “non-medical caregivers” in the chapter “2.2. Sample size”.

3. Quetions of the questionnaire seem more general to vaccination and not to cocoon strategy: please clarify

Response: Yes, we added an explanation in the chapter “2.1 Data”:

[The questionnaire] comprised general statements about vaccination without introducing the concept of the cocoon strategy to avoid bias.

4. The population selected has some subtypes which may necessitate more sample size.

Response: Yes. For this survey, the sample size was adequate to reach statistically relevant results and conclusions. However, we planned consecutive research focusing on selected professional subtypes, comprising at least 400 respondents in each group.

Reviewer 2 Report

The concept of a cocoon approach to protecting high risk subjects is important. The introduction is largely irrelevant to this issue, and needs significant shortening and focussing on the central question. The methodology requires better description of statistical methods, which seem to have far too greater significance (eg P,0.0001 etc for differences in small numbers of 20-30%). It would be helpful to have a statistician review these data. The important contribution is the low vaccination rates of health and support workers involved in very high risk groups - it would be valuable to have comparative data for Polish Aged care, where longer term subjects would be exposed. The paper should be shortened and focussed on the single issue addressed.

This paper addresses one significant issue, yet the introduction , discussion etc is poorly focussed, and needs re-writing. Someone more familiar with English should edit the paper. The statistical analysis needs clarification, as the P values quoted seem disproportionally low for numbers and size differences. Comparison with immunisation rates in other professional and support groups in Poland would be helpful. Remedial activity could be better defined.

Author Response

Thank you for your invaluable remarks and suggestions. We followed and implemented them thoroughly. Please, find our responses below in consecutive order.

The concept of a cocoon approach to protecting high risk subjects is important. The introduction is largely irrelevant to this issue, and needs significant shortening and focussing on the central question.

Response: We fully agree. We have shortened the introduction significantly, focusing  on on the paper’s goal.

The methodology requires better description of statistical methods, which seem to have far too greater significance (eg P,0.0001 etc for differences in small numbers of 20-30%). It would be helpful to have a statistician review these data.

Response: We believe that P = 0,05 is adequate for the survey. Please, note that we deployed Bonferroni correction for multiple comparisons, as stated in the “2.3 Statistical analysis” chapter. It means that alpha = 0.0019 was used where appropriate and in each case it has been underlined consistently.

The important contribution is the low vaccination rates of health and support workers involved in very high risk groups - it would be valuable to have comparative data for Polish Aged care, where longer term subjects would be exposed.

Response: Unfortunately, there is no data for the older adults population. We are aware of some small sample size of pregnant women, which is irrelevant for our purpose. However, we planned further research focused on the geriatric population.

The paper should be shortened and focussed on the single issue addressed.

Response: We agree. We shortened the introduction more than by half and focused only on the goal of our paper. The results are relevant for our next steps in the research.

This paper addresses one significant issue, yet the introduction , discussion etc is poorly focussed, and needs re-writing. Someone more familiar with English should edit the paper.

Response: Sadly, there were numerous grammar, punctuation, and wording errors. We thoroughly reviewed and proofed the paper.

The statistical analysis needs clarification, as the P values quoted seem disproportionally low for numbers and size differences.

Response: We hardly agree with this comment. All results are thoroughly checked. We are ready to present you with the raw data for verification.

Comparison with immunisation rates in other professional and support groups in Poland would be helpful.

Response: Yes, it would be helpful. Unfortunately, our survey is the first and only one performed in Poland so far. We do not have such data in Poland. We can compare for example nurses in Poland to nurses in Germany, but not nurses in Poland to physiotherapists in Germany, which may be misleading.

Remedial activity could be better defined.

Response: Good point. We added the last sentence. We planned a consecutive research to define the remedial activity. In fact this is the logical consequence of this survey.

Round 2

Reviewer 2 Report

I think the paper is marginal but acceptable for publication.

 Someone more familiar with English should edit the paper.